# Infant HIV-protection: Comparing antiretroviral therapy, maternal and infant factors influence on infant HIV acquisition in Uganda: A six-year real-world experience

**Collins Ankunda**[1,2,3]*, **Jude Emunyu**[4], **Brendah Kyomuhangi**[5], **Sharon Namasambi**[6], **Conrad Sserunjogi**[3], **Iving Mumbere**[7], **Jane Nakaweesi**[8]

**1** Department of Pharmacology and Therapeutics, Makerere University, Kampala Uganda, **2** Department of Global Public health, Karolinska Institute, Stockholm, Sweden, **3** Mildmay Uganda Research Centre, Naziba Hill, Kampala, Uganda, **4** Makerere University Walter Reed Program, Kampala, Uganda, **5** Entebbe Regional Referral Hospital, Entebbe, Uganda, **6** Uganda Public Health Fellowship Program, Uganda National Institute of Public Health, Kampala, Uganda, **7** School of Public Health, Makerere University, Kampala, Uganda, **8** Baylor College of Medicine Children's Foundation, Kampala, Uganda

* ankundacollins@gmail.com

## Abstract

Although antiretroviral therapy (ART) scale up has markedly lowered vertical transmission, the relative contribution of ART regimens, maternal factors, and infant practices within Prevention of vertical transmission (PVT) programs is not fully understood. We assessed the comparative influence of maternal and infant factors on infant HIV positivity. We retrospectively reviewed records of 962 HIV exposed infants attending the Mildmay Uganda PVT clinic between 2018 and 2023. Of these, 918 were mother-infant pairs. Descriptive statistics summarised maternal and infant characteristics. Associations were assessed using Chi square or Fisher exact tests. Two independent logistic regression models identified infant and maternal factors associated with HIV positivity. Pseudo R² evaluated model performance and area under the curve measured discrimination. Among 918 mother-infant pairs, HIV positivity was 1.96% (18). PI based regimens were associated with increased odds of HIV positivity (aOR 22.89, 95% CI 2.20–238.14, p = 0.009). Among maternal factors, HIV positivity differed significantly by age group (p < 0.001) and viral load suppression status (p = 0.025), with viral load non-suppression independently increasing risk (aOR 4.97, 95% CI 1.45–17.10, p = 0.011). Among infant factors, mixed feeding (aOR 26.87, 95% CI 5.82–124.10) and no breastfeeding (aOR 86.94, 95%CI 17.40–434.56) were strongly associated with HIV positivity (both p < 0.001). The maternal model showed modest explanatory capacity (pseudo R² 0.09) and acceptable discrimination (AUC 0.74; bootstrap AUC 0.70, 95% CI 0.58–0.83). The infant model demonstrated stronger performance (pseudo R² 0.51) with excellent discrimination (AUC 0.94; cross-validated AUC 0.95, 95% CI 0.79–0.95).While maternal ART regimen and viral suppression are critical, infant feeding practices appear to have a stronger influence

**Data availability statement:** All data used for this manuscript are now included within the manuscript itself and the supplementary materials.

**Funding:** This work was supported by the National Institute for Health and Care Research (NIHR) through the Royal Society for Tropical Medicine and Hygiene Early Career Grants Programme (to CA). The funder of the study had no role in study design, data collection, data analysis, data interpretation, writing of the report, or the decision to submit this manuscript for publication.

**Competing interests:** The authors have declared that no competing interests exist.

on infant HIV positivity. Strengthening counselling on optimal infant feeding practices and timely HIV prophylaxis is essential to achieve elimination of paediatric HIV.

## Background

Globally, vertical transmission (VT) remains a major source of paediatric HIV infection, particularly in sub-Saharan Africa [1], where the burden of HIV is high. However, ART scale-up among pregnant women living with HIV (PWLH) has significantly reduced transmission risk during pregnancy, delivery, and breastfeeding. Without intervention, VT risk ranges from 15% to 45% [1], but with effective ART and comprehensive PVT measures, it can be reduced to under 5%, or even below 2% in some settings [1–3].

Several maternal factors influence the risk of vertical transmission during pregnancy and the postpartum period, including ART use, maternal viral load, overall health status, breastfeeding practices, and coexisting sexually transmitted infections (STIs), such as syphilis [4,5]. Among these, a high maternal viral load during pregnancy, delivery, or breastfeeding is a key driver of transmission [6]. Effective antiretroviral therapy (ART) regimens that achieve rapid and sustained viral load suppression are essential for preventing VT [7,8]. This highlights the importance of timely ART initiation, sustained adherence, and robust maternal healthcare systems, with a preference for regimens that reliably and quickly suppress viral replication [9,10]. Dolutegravir (DTG)-based regimens have become the preferred first-line treatment for PWLH due to their superior efficacy in rapidly suppressing viral load compared to previous regimens, such as efavirenz (EFV)-based combinations [2,11,12].

Infant feeding practices and maternal viral load are major determinants of postnatal vertical transmission risk [13,14]. Accordingly, breastfeeding plays a pivotal role in shaping postpartum transmission outcomes. The World Health Organization recommends exclusive breastfeeding for the first six months of life, followed by the introduction of appropriate complementary foods while continuing breastfeeding up to one year or beyond, contingent on sustained maternal viral suppression [15]. In parallel, infant antiretroviral prophylaxis, such as nevirapine, is tailored according to the infant's level of HIV exposure risk [2]. Maintaining sustained maternal viral load suppression throughout the breastfeeding period remains critical to minimizing postnatal HIV transmission among HIV-exposed infants [10].

Although the effectiveness of postnatal PVT interventions is well established, existing studies largely assess the overall impact of ART and infant prophylaxis without disentangling the relative contributions of antepartum and postpartum interventions. Consequently, there is limited evidence directly comparing how interventions at different stages of the maternal–infant care continuum influence infant HIV outcomes. The study compared the impact of antepartum and postpartum factors that influence infant HIV positive status among PWLH on different ART regimens. By examining these factors across the continuum of care, the study aims to inform strategies to optimise PVT outcomes and advance progress toward the elimination of paediatric HIV in high-burden settings.

## Methods and materials

### Ethics statement

Ethical approval was obtained from the Mildmay Uganda Research Ethics Committee (#REC REF 0201–2024) and the Uganda National Council for Science and Technology (HS3873ES). A waiver of informed consent was granted by Mildmay Uganda Research Ethics Committee for the use of de-identified retrospective data. Administrative clearance was sought from MUgH administration to access participant data.

### Study design

This was a retrospective cohort study that reviewed medical records of PWLH enrolled in the PVT clinic at the Mildmay Uganda Hospital (MUgH) from 2018 to 2023.

### Study site and population

This study was conducted at MUgH, a peri-urban facility in Wakiso District with 50-bed capacity, providing integrated TB-HIV care and ART to nearly 15,000 active HIV patients [16]. The study population included HIV-exposed infants with documented final HIV test results in accordance with Ministry of Health guidelines. Infant HIV diagnosis in Uganda is performed using DNA PCR for infants under 18 months and serology for infants aged 18–24 months [2].

### Data collection and management

Data collection and participant recruitment were conducted over a three-month period (1st July to 30th September 2024). Data on participants' socio-demographic characteristics and medical history were extracted from PVT registers, patient cards, and Electronic Medical Records (EMR). Age was categorized into three groups;15–24, 25–34, and >35 years based on biological relevance and previous research [17]. HIV severity was classified according to WHO stages: Stage 1 (asymptomatic), Stage 2 (mild symptoms), Stage 3 (advanced disease), and Stage 4 (severe/AIDS-defining illness). Viral load suppression was defined as HIV RNA ≤ 200 copies/mL, while non-suppression >200 copies/mL, in line with WHO guidelines [2,3]. Drug lines were classified per WHO recommendations: first line ART refers to the initial regimen for starting therapy, second line is used after first line failure, and third line is reserved for patients failing both first and second line regimens [3]. Data entry was performed using Microsoft Excel, with regular accuracy checks against hard copy records to ensure data integrity.

A flowchart illustrates the data collection process as shown in Fig 1 below.

### Statistical analysis plan

Data were analysed using Stata version 17 (StataCorp, College Station, TX, USA). The primary outcome was infant HIV status, defined as a binary variable (positive vs. negative). Maternal and infant characteristics were summarized using frequencies and percentages, with group differences assessed using chi-square or Fisher's exact tests, as appropriate. Differences between maternal ART regimen and infant HIV status were evaluated using Fisher's exact test and were restricted to linked mother–infant pairs. Two separate multivariable logistic regression models were fitted: one evaluating maternal factors and a second independently evaluating infant factors. Crude odds ratios (COR) and adjusted odds ratios (aOR), with 95% confidence intervals (CI), were reported with statistical significance defined as $p < 0.05$. Variables associated with infant HIV positivity at $p < 0.20$ in bivariate logistic regression, together with biologically plausible factors, were considered for multivariable analysis [18].

Evaluation of model performance was determined using Pseudo $R^2$ and discrimination metrics of the area under the receiver operating characteristic curve (AUC). Predictive robustness was assessed using cross-validated AUC. However,

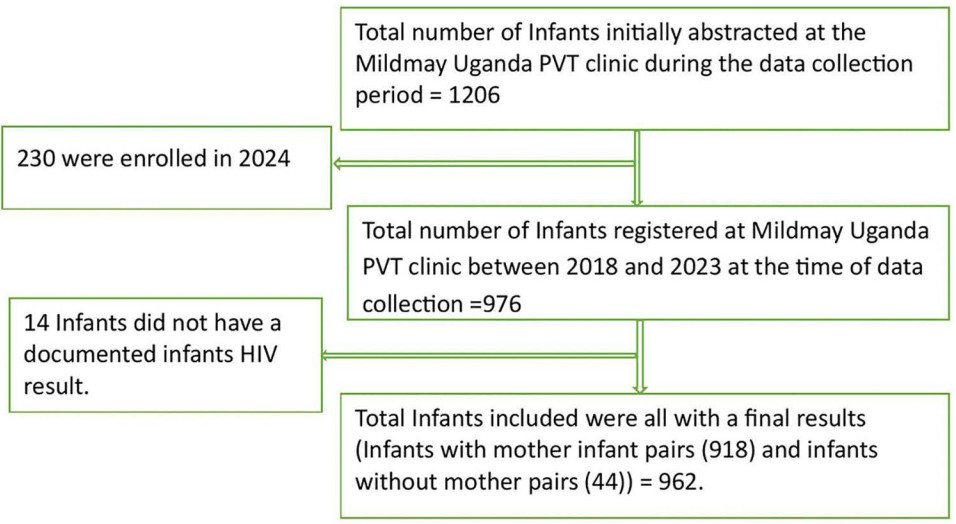

**Fig 1. A flowchart.**

due to sparse outcome events which precluded reliable cross-validation in the maternal model, predictive robustness was evaluated using bootstrap internal validation AUC. Formal calibration metrics were not applied due to the very small number of outcome events, which limit their reliability. Given the low level of outcome missingness (1.4%; 14/976), exclusion was unlikely to materially affect study validity or power. Predictor variables with missing data were retained by coding them as "unknown" or "not sure" to minimize potential bias and preserve sample size.

Collinearity among ART-related variables (regimen, duration, and prophylaxis) was assessed using variance inflation factors, with values <5 indicating no problematic multicollinearity.

Sensitivity analyses using Firth's penalized logistic regression were conducted to address sparse outcome events and yielded directionally consistent estimates with comparable confidence intervals.

A post hoc power analysis was performed using simulation to estimate the study's ability to detect associations of clinically meaningful magnitude. Assuming an odds ratio of 6, reflecting moderate maternal effects and stronger infant effects, a sample size of 962, and a baseline event rate of 1.96%, 2,000 simulated datasets were generated under a logistic model. Power was estimated as the proportion of simulations yielding $p < 0.05$ and was 83%, indicating adequate sensitivity to detect effects of this magnitude despite the low outcome prevalence.

## Results

### Distribution of infant HIV positivity by maternal and infant characteristics

Among the 918 mother-infant pairs, key maternal factors significantly associated with infant HIV positivity included maternal age ($p < 0.001$), with the highest positivity among women over 35 years (5.8%), and PWLH with a non-suppressed viral load (6.7% vs. 1.6%, $p = 0.025$). When all the 962 infants were assessed, all infant-related factors were significantly associated with infant HIV positivity. Infants not receiving nevirapine prophylaxis had a markedly higher HIV positivity rate (53.7% vs. 2.3%, $p < 0.001$). Similarly, those not on cotrimoxazole had a 50% positivity rate compared to 4.1% among those who received it ($p < 0.001$). Feeding practices also impacted outcomes: exclusive breastfeeding had the lowest HIV positivity (0.4%), while mixed and no breastfeeding had significantly higher rates (9.1% and 25.0%, respectively; $p < 0.001$) (Table 1).

**Table 1. Distribution of infant HIV status by maternal and infant characteristics.**

| Group Characteristics | Variable | Categories | HIV Negative | HIV Positive | P-value** |
|---|---|---|---|---|---|
| **Maternal Characteristics** | Age (Years) | 15-24 | 114 (100.0) | 0 (0.0) | **<0.001** |
| | | 25-34 | 510 (99.8) | 1 (0.2) | |
| | | >35 | 276 (94.2) | 17 (5.8) | |
| | WHO stage | Stage 1 | 847 (97.9) | 18 (2.1) | 1.000 |
| | | Stage 2 | 19 (100.0) | 0 (0.0) | |
| | | Stage 3 | 9 (100.0) | 0 (0.0) | |
| | | Stage 4 | – | – | |
| | Adherence | Good (≥95%) | 889 (98.1) | 17 (1.9) | 0.181 |
| | | Fair (85–94%) | 2 (100.0) | 0 (0.0) | |
| | | Poor (<85%) | 7 (87.5) | 1 (12.5) | |
| | Start Regimen Anchor | Dolutegravir (DTG) | 119 (98.4) | 2 (1.6) | 0.550 |
| | | Efavirenz (EFV) | 605 (98.2) | 11 (1.8) | |
| | | Nevirapine (NVP) | 143 (98.0) | 3 (2) | |
| | | Protease Inhibitor(PI) | 19 (95.0) | 1 (5.0) | |
| | Drug line | First line | 819 (98.1) | 16 (1.9) | 0.678 |
| | | Second line | 80 (97.6) | 2 (2.4) | |
| | | Third line | 1 (100.0) | 0 (0.0) | |
| | Duration on ART (years) | >1 | 26 (92.9) | 2 (7.1) | 0.114 |
| | | 1-2 | 73 (97.3) | 2 (2.7) | |
| | | >2-5 | 243 (97.6) | 6 (2.4) | |
| | | >5 | 558 (98.6) | 8 (1.4) | |
| | History of regimen change | No | 109 (98.2) | 2 (1.8) | 1.000 |
| | | Yes | 791 (98.0) | 16 (2.0) | |
| | ART experience | Experienced | 872 (98.2) | 16 (1.8) | 0.114 |
| | | Naive | 28 (93.3) | 2 (6.7) | |
| | Viral load suppression during pregnancy | Suppressed | 844 (98.4) | 14 (1.6) | **0.025** |
| | | Unsuppressed | 56 (93.3) | 4 (6.7) | |
| | Marital status | Married | 433 (97.7) | 10 (2.3) | 0.854 |
| | | Single | 303 (98.4) | 5 (1.6) | |
| | | Separated/Divorced/Widowed | 79 (97.5) | 2 (2.5) | |
| | | Unknown | 85 (98.8) | 1 (1.2) | |
| | Mode of delivery | Caesarean section | 242 (98.4) | 4 (1.6) | 1.000 |
| | | Spontaneous Vaginal Delivery (SVD) | 637 (98.0) | 13 (2.0) | |
| | Place of delivery | Health facility | 896 (98.0) | 18 (2.0) | 1.000 |
| | | Home/ Unknown | 22 (100.0) | 0 (0.0) | |
| | History of ART transition* | 0 | 448 (98.9) | 5 (1.1) | 0.112 |
| | | 1 | 19 (100.0) | 0 (0.0) | |
| | | 2 | 27 (96.4) | 1 (3.6) | |
| | | 3 | 3 (100.0) | 0 (0.0) | |
| | | 4 | 107 (98.2) | 2 (1.8) | |
| | | 5 | 237 (96.7) | 8 (3.3) | |
| | | 6 | 5 (83.3) | 1 (16.7) | |
| | | 7 | 40 (100.0) | 0 (0.0) | |

*(Continued)*

**Table 1.** (Continued)

| Group Characteristics | Variable | Categories | HIV Negative | HIV Positive | P-value** |
|---|---|---|---|---|---|
| **Infant Characteristics** | Received NVP syrup | Yes | 900 (97.7) | 21 (2.3) | **p<0.001** |
| | | No/Not sure | 19 (46.3) | 22 (53.7) | |
| | Received Cotrimoxazole prophylaxis | Yes | 915 (95.9) | 39 (100.0) | **p<0.001** |
| | | No/not sure | 4 (50.0) | 4 (50.0) | |
| | Infant Feeding option at the time of first test | Exclusive Breast feeding | 806 (99.6) | 3 (0.4) | **p<0.001** |
| | | Mixed feeding | 40 (90.9) | 4 (9.1) | |
| | | Not breast feeding | 12 (75.0) | 4 (25.0) | |
| | | Replacement feeding | 35 (97.2) | 1 (2.8) | |
| | | Unknown | 26 (45.6) | 31 (54.4) | |

* The study categorized antiretroviral therapy (ART) exposure based on maternal regimen transitions during pregnancy and breastfeeding. These included: transition from other regimens to DTG-based (0), EFV-based (1), PI-based (2), or NVP-based (3) regimens, as well as cases where PWLH remained on the same regimen throughout; DTG-based (4), EFV-based (5), PI-based (6), or NVP-based (7).

** P-values were derived using Chi-square tests, or Fisher's exact tests when cell counts were <5.

## Influence of PVT ART regimen on infant HIV positivity

Among 918 HIV-exposed infants with mother pairs, 18 (1.96%, 95% CI: 1.15–3.06) were confirmed HIV positive. When stratified by maternal ART regimen, HIV positivity was 1.40% (8/571; 95% CI: 0.61–2.74) for infants whose mothers were on DTG-based regimens, 2.96% (8/270; 95% CI: 1.29–5.75) for those on Efavirenz(EFV)-based regimens, 0% (0/43; 95% CI: 0.00–8.22) for Nevirapine(NVP)-based regimens, and 5.88% (2/34; 95% CI: 0.72–19.68) for Protease inhibitor(PI)–based regimens. The difference in HIV positivity rates across these ART regimens was not statistically significant (p=0.113) as shown in Table 2 below.

## Bivariate and multivariate analysis of maternal and infant factors associated with VT of HIV

Among maternal factors, being on a PI-based regimen showed higher odds of VT compared with DTG (COR=4.32; 95% CI: 0.88–21.18; p=0.071), with a stronger association observed in multivariable analysis (aOR = 22.89; 95% CI: 2.20–238.14; p=0.009). Maternal age 25–34 years was associated with significantly lower odds of VT compared with mothers aged >35 years (COR=0.03; 95% CI: 0.00–0.24; p=0.001) though the association was not sustained after adjustment. Unsuppressed viral load was independently associated with increased VT risk (aOR = 4.97; 95% CI: 1.45–17.10; p=0.011).

Among infant factors, lack of nevirapine prophylaxis or uncertain receipt was strongly associated with higher odds of VT in bivariate analysis (COR=49.62; 95% CI: 23.42–105.16; p<0.001); however, this association was not retained after adjustment (aOR = 1.25; 95% CI: 0.43–3.69; p=0.682). Similarly, lack of cotrimoxazole prophylaxis was associated with increased VT risk in bivariate analysis (COR=23.46; 95% CI: 5.66–97.30; p<0.001), but not in the multivariable model

**Table 2.** Proportion of HIV positive infants by PVT ART regimen category.

| PMTCT ART Regimen | Number of Infants | Frequency of HIV positivity | HIV positivity % (95% CI) | Fisher's exact p-value |
|---|---|---|---|---|
| DTG BASED | 571 | 8 | 1.40 (0.61–2.74) | 0.113 |
| EFV BASED | 270 | 8 | 2.96 (1.29–5.75) | |
| NVP BASED | 43 | 0 | 0.00 (0.00–8.22) | |
| PI BASED | 34 | 2 | 5.88 (0.72–19.68) | |
| Total | 918 | 18 | 1.96 (1.15–3.06) | |

(aOR = 1.02; 95% CI: 0.20–5.22; p = 0.983). In contrast, infant feeding practices remained strongly associated with VT after adjustment. Compared with exclusive breastfeeding, mixed feeding (aOR = 26.87; 95% CI: 5.82–124.10; p < 0.001) and not breastfeeding (aOR = 86.94; 95% CI: 17.40–434.56; p < 0.001) were independently associated with higher odds of VT (Table 3).

## Comparison of Maternal and Infant Predictors of VT of HIV

The maternal factors model (N = 821) yielded a log-likelihood of −74.54, a pseudo-R² of 0.09, and 10 degrees of freedom, with a standard area under the receiver operating characteristic curve (AUC) of 0.7. Since cross-validated AUC was unreliable due to sparse events, so bootstrap resampling was used for internal validation, yielding moderate discrimination (AUC 0.70, bootstrap 95% CI 0.58–0.83).

In contrast, the infant factors model (N = 962) demonstrated a log-likelihood of −85.95, a pseudo-R² of 0.51, and 7 degrees of freedom, with a standard AUC of 0.94 and a cross-validated AUC of 0.95 (95% CI: 0.79–0.95). AIC and BIC values for each model are reported descriptively as shown in Table 4 below.

## Discussion

This study comprehensively assessed the relative contributions of maternal and infant factors to vertical transmission (VT) of HIV in Uganda. With an overall infant HIV positivity of 1.96% across maternal ART regimens, the findings emphasising the substantial progress achieved through widespread PVT interventions [1–3], while highlighting persistent challenges in the care of HIV-exposed infants. Additionally, no significant difference in HIV positivity was observed across ART regimens (p = 0.113). However, both bivariable and multivariable analyses indicated higher odds of HIV positivity with PI-based regimens, likely reflecting adherence challenges due to higher pill burden and twice-daily dosing. Programmatically, PIs are typically prescribed as second-line therapy, often among individuals with prior treatment failure, which may further confound associations [12,19]. Overall, DTG-, EFV-, and NVP-based regimens appear to provide comparable protection against mother-to-child HIV transmission [20].

Differences in HIV positivity across maternal age groups were significant (p < 0.001). HIV positivity occurred predominantly among mothers aged ≥35 years, while younger age groups showed none or very few events, resulting in perfect prediction in regression analysis. Similarly, a study done in India reported that women aged 30 years and above had a sevenfold increased risk of VT [19]. This finding may reflect HIV program risk stratification, as younger mothers are often perceived as higher risk and thus receive enhanced support within PVT programs, while older mothers may receive relatively less intensive intervention. While previous literature often associates older age with better antiretroviral therapy (ART) outcomes due to presumed greater maturity, awareness, and adherence [17,18], this trend was not observed in our cohort. Furthermore, our study showed that unsuppressed maternal viral load was associated with HIV positivity, consistent with the established link between maternal viremia and infant infection [2,21,22]. This emphasises viral load suppression as a cornerstone of PVT success highlighting the need for routine viral load monitoring among pregnant and breast-feeding women for better exposed HIV infant outcomes [2].

The noted statistical differences in HIV positivity across infant-related factors emphasizes their critical role in prevention of postnatal HIV transmission. However, in the regression analysis, infant feeding practices emerged as a strong factor associated with HIV positivity, with exclusive breastfeeding associated with the lowest HIV positivity. These findings are consistent with biological mechanisms: mixed feeding can damage the infant gut mucosa, facilitating viral entry, while exclusive breastfeeding maintains gut integrity and transmits protective maternal antibodies [23]. The heightened risk in infants not breastfed may relate to lack of immune protection and replacement feeding challenges in resource-limited settings [2,23]. The high odds to HIV positivity among infants with unknown feeding practices may highlight data gaps and possible non-adherence to recommended feeding guidelines.

**Table 3. Bivariate and multivariate analysis of maternal and infant factors associated with VT of HIV.**

| Models | Variable | Categories | COR (95% CI) | p-value | aOR (95% CI) | P-Value |
|---|---|---|---|---|---|---|
| **Maternal factors (N=918)** | ART Regimen | DTG | Ref | | Ref | |
| | | EFV | 2.19 (0.81–5.91) | 0.120 | 2.42 (0.74–7.87) | 0.142 |
| | | PI | 4.32 (0.88–21.18) | **0.071** | 22.89 (2.20–238.14) | **0.009** |
| | | NVP | 1 | – | | |
| | Age (Years) | 15–24 | 1 | – | | |
| | | 25–34 | 0.03 (0.00–0.24) | **0.001** | – | – |
| | | >35 | Ref | | | |
| | Adherence | Good | Ref | | | |
| | | Fair | 1 | – | | |
| | | Poor | 7.47 (0.87–64.10) | 0.067 | – | – |
| | Regimen at ART initiation | DTG | Ref | | | |
| | | EFV | 1.08 (0.24–4.94) | 0.919 | | |
| | | PI | 3.13 (0.27–36.24) | 0.361 | | |
| | | NVP | 1.25 (0.21–7.59) | 0.810 | | |
| | Drug line during PTCMT | First line | Ref | | | |
| | | Second line | 1.31 (0.30–5.80) | 0.721 | | |
| | | Third line | 1 | – | | |
| | Duration on ART (years) | >1 | 3.12 (0.60–16.23) | 0.177 | 4.63 (0.73–29.29) | 0.104 |
| | | 1–2 | 1.11 (0.22–5.62) | 0.900 | 1.22 (0.19–7.76) | 0.835 |
| | | >2–5 | Ref | | Ref | |
| | | >5 | 0.58 (0.20–1.69) | 0.319 | 0.61 (0.19–2.02) | 0.421 |
| | History of regimen change | No | Ref | | | |
| | | Yes | 1.09 (0.25–4.82) | 0.906 | | |
| | ART experience | Experienced | Ref | | | |
| | | Naive | 3.89 (0.85–17.75) | 0.079 | – | |
| | Viral load suppression | Suppressed | Ref | | Ref | |
| | | Unsuppressed | 3.89 (0.85–17.75) | 0.079 | 4.97 (1.45–17.10) | **0.011** |
| | Marital status | Married | Ref | | | |
| | | Single | 0.71 (0.24–2.11) | 0.543 | | |
| | | Separated/Divorced/ Widowed | 1.10 (0.24–5.10) | 0.907 | | |
| | | Unknown | 0.51 (0.06–4.03) | 0.523 | | |
| | Mode of delivery | Caesarean section | Ref | | | |
| | | SVD | 1.23 (0.40–3.82) | 0.715 | | |
| | ART history of transition* | 0 | Ref | | Ref | |
| | | 1 | 1 | – | 1 | – |
| | | 2 | 3.32 (0.37–29.41) | 0.281 | 0.14 (0.01–2.78) | 0.198 |
| | | 3 | 1 | – | 1 | – |
| | | 4 | 1.67 (0.32–8.75) | 0.541 | 0.80 (0.11–5.62) | 0.821 |
| | | 5 | 3.02 (0.98–9.35) | 0.055 | 1 | – |
| | | 6 | 17.92 (1.76–182.51) | 0.015 | 1 | – |
| | | 7 | 1 | – | 1 | – |

*(Continued)*

**Table 3.** (Continued)

| Models | Variable | Categories | COR (95% CI) | p-value | aOR (95% CI) | P-Value |
|--------|----------|-----------|--------------|---------|--------------|---------|
| **Infant factors (N = 962)** | Received NVP syrup | Yes | Ref | | Ref | |
| | | No/Not sure | 49.62 (23.42–105.16) | **<0.001** | 1.25 (0.43–3.69) | 0.682 |
| | Received Cotrimoxazole prophylaxis | Yes | Ref | | Ref | |
| | | No/not sure | 23.46 (5.66–97–30) | **<0.001** | 1.02 (0.20–5.22) | 0.983 |
| | Infant Feeding option at the time of first test | Exclusive Breast feeding | Ref | | Ref | |
| | | Mixed feeding | 26.87 (5.82–124.11) | **<0.001** | 26.87 (5.82–124.10) | **<0.001** |
| | | Not breast feeding | 89.56 (18.05–444.37) | **<0.001** | 86.94 (17.40–434.56) | **<0.001** |
| | | Replacement feeding | 7.68 (0.78–75.68) | 0.081 | 7.62 (0.77–75.16) | 0.082 |
| | | Unknown | 320.33 (91.98–1115.58) | **<0.001** | 275 (66.04–1145.72) | **<0.001** |

* The study categorized antiretroviral therapy (ART) exposure based on maternal regimen transitions during pregnancy and breastfeeding. These included: transition from other regimens to DTG-based (0), EFV-based (1), PI-based (2), or NVP-based (3) regimens, as well as cases where PWLH remained on the same regimen throughout; DTG-based (4), EFV-based (5), PI-based (6), or NVP-based (7).

**Table 4. Model fit statistics for maternal and infant predictors of VT of HIV.**

| Model | Number of observations | Log Likelihood | Pseudo R² | Degrees of freedom | AIC | BIC | Standard AUC | Cross validated AUC(95% CI) |
|-------|------------------------|----------------|-----------|--------------------|-----|-----|--------------|------------------------------|
| **Maternal** | 821 | -74.54 | 0.09 | 10 | 160.09 | 207.19 | 0.74 | 0.70 (0.58–0.83)* |
| **Infant** | 962 | -85.95 | 0.51 | 7 | 185.91 | 219.99 | 0.94 | 0.95 (0.79-0.95) |

* Bootstrap resampling 95% confidence interval; not cross-validated.

This study compared maternal and infant logistic regression models to evaluate predictors of infant HIV positivity. The maternal model showed modest explanatory power (pseudo-R² = 0.09) and acceptable discrimination (AUC = 0.74), reflecting limited ability of maternal factors alone to distinguish HIV-positive from HIV-negative infants. Sparse outcome events further prevented reliable cross-validated AUC estimation, highlighting reduced maternal model stability. In contrast, the infant model demonstrated substantially stronger performance (pseudo-R² = 0.51) with excellent discrimination (AUC = 0.94; cross-validated AUC = 0.95, 95% CI 0.79–0.95), indicating robust predictive capacity. These findings suggest that infant-level factors provide greater explanatory and discriminatory value than maternal factors alone, highlighting the dual importance of maternal and infant interventions; as maternal factors reduce intrauterine and intrapartum transmission, while infant care practices drive postnatal HIV positivity. Maintaining maternal viral suppression and promoting safe infant care practices, including exclusive breastfeeding, are therefore essential in PVT programs [2,24].

## Implications for policy and practice

Our findings highlight the importance of scaling up all ART regimens in PVT programs, except PI-based regimens, unless DTG-, EFV-, or NVP-based regimens are contraindicated or have documented resistance. Policy should avoid unnecessary regimen switch or transition among pregnant and breast-feeding women unless clinically indicated.

In addition, special attention is warranted for older mothers and those with unsuppressed viral loads, who represent a higher-risk group. Interventions such as targeted adherence counselling, peer to peer psychosocial support, viral load monitoring, and regimen optimization could reduce their transmission risk.

Importantly, during the postpartum period, infant prophylaxis coverage, timely initiation of Nevirapine and cotrimoxazole for all HIV-exposed infants, and exclusive breastfeeding promotion are key actionable areas in PVT programming.

### Strengths and limitations

This study's strengths include a large sample size and clinic-based data collected over six years, offering a comprehensive evaluation of both maternal and infant factors within a real-world clinical context. This enhances the generalizability of the findings. However, some limitations must be acknowledged. The inherent limitations of a retrospective design, including the inability to infer causality and potential bias from incomplete data, may have affected the accuracy of some point estimates. Small numbers in some ART regimen subgroups reduced statistical power, and self-reported infant feeding practices may be subject to recall or social desirability bias. In addition, the low number of outcome events may have increased uncertainty in effect estimates; however, Sensitivity analysis using Firth's penalised logistic regression evaluated robustness, producing directionally consistent estimates with comparable confidence intervals despite sparse data. Additionally, unmeasured factors such as maternal nutritional status, co-infections, socio-economic conditions, and adherence support interventions could confound the observed associations. Despite these constraints, the study provides important insights into the relative contributions of maternal and infant factors to VT and identifies key areas for strengthening the PVT program.

### Conclusion

While maternal factors such as ART regimen and viral load suppression appear important in reducing VT risk, infant factors, particularly prophylaxis and feeding practices, may play a more influential role in shaping infant HIV positivity. These findings emphasize the need for integrated approaches that concurrently optimize maternal treatment and infant care to achieve the goal of eliminating paediatric HIV.

Targeted strategies promoting exclusive breastfeeding, ensuring prophylaxis coverage, and supporting maternal viral suppression, especially among older women, are essential. Strengthening health systems to improve postnatal follow-up will further augment PVT programs. Continued research focusing on disentangling postnatal transmission pathways, including adherence to exclusive breastfeeding, and transitions in feeding practices remain critical in sustaining and accelerating progress toward an HIV-free generation in Uganda and beyond.

### Supporting information

**S1 Fig. Impact of negative maternal and infant factors on infant HIV positive outcome.**
(DOCX)

**S2 Fig. HIV testing algorithm for HIV exposed infants below 18 months.**
(TIF)

**S1 Table. Showing time of positive HIV diagnosis among infants.**
(DOCX)

**S2 Table. Maternal characteristics for positive infants.**
(DOCX)

**S3 Table. Sensitivity analysis-penalized regression.**
(DOCX)

**S1 Data. Full study dataset.**
(XLSX)

## Acknowledgments

The authors sincerely thank the study participants, hospital administrators, staff, and healthcare workers, whose contributions at every level were essential to the success of this study. We also acknowledge the support of Ps. Philip Omondi throughout the study. In addition, we recognize the valuable contributions of the Intent Health Research Group (https://intentresearchgroup.com).

## Author contributions

**Conceptualization:** Collins Ankunda, Jude Emunyu, Brendah Kyomuhangi, Sharon Namasambi, Conrad Sserunjogi, Iving Mumbere, Jane Nakaweesi.

**Data curation:** Collins Ankunda, Jude Emunyu, Brendah Kyomuhangi, Sharon Namasambi, Conrad Sserunjogi, Iving Mumbere, Jane Nakaweesi.

**Formal analysis:** Collins Ankunda, Jude Emunyu, Brendah Kyomuhangi.

**Funding acquisition:** Collins Ankunda.

**Investigation:** Collins Ankunda, Jude Emunyu, Brendah Kyomuhangi, Sharon Namasambi, Jane Nakaweesi.

**Methodology:** Collins Ankunda, Jude Emunyu, Brendah Kyomuhangi, Sharon Namasambi, Conrad Sserunjogi, Iving Mumbere, Jane Nakaweesi.

**Project administration:** Collins Ankunda, Jane Nakaweesi.

**Resources:** Collins Ankunda.

**Supervision:** Collins Ankunda.

**Validation:** Jude Emunyu, Conrad Sserunjogi.

**Writing – original draft:** Collins Ankunda, Jude Emunyu, Brendah Kyomuhangi, Sharon Namasambi, Conrad Sserunjogi, Iving Mumbere, Jane Nakaweesi.

**Writing – review & editing:** Collins Ankunda, Jude Emunyu, Brendah Kyomuhangi, Sharon Namasambi, Conrad Sserunjogi, Iving Mumbere, Jane Nakaweesi.

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
