## [Decision Letter · Decision Letter 0]

7 Oct 2025

PGPH-D-25-01533

Infant Sero-protection: Comparing Antiretroviral therapy, Maternal and Infant Factors Influence on Infant HIV acquisition in Uganda: A six-year Real-World Perspective.

Dear Dr. Ankunda,

Thank you for submitting your manuscript to PLOS Global Public Health. After careful consideration, we feel that it has merit but does not fully meet PLOS Global Public Health’s publication criteria as it currently stands. Therefore, we invite you to submit a revised version of the manuscript that addresses the points raised during the review process.

We look forward to receiving your revised manuscript.

Kind regards,

Nisha Anne Sunny Jacob, MBChB, M.Med, FCPHM (SA), PhD

Guest Editor

Journal Requirements:

1. Please send a completed 'Competing Interests' statement, including any COIs declared by your co-authors. If you have no competing interests to declare, please state "The authors have declared that no competing interests exist". Otherwise please declare all competing interests beginning with the statement "I have read the journal's policy and the authors of this manuscript have the following competing interests:"

1. State the initials, alongside each funding source, of each author to receive each grant. For example: "This work was supported by the National Institutes of Health (####### to AM; ###### to CJ) and the National Science Foundation (###### to AM)."

3. In the online submission form, you indicated that “The data sets for this study are available upon reasonable request from the corresponding author.”

3. Uploaded as supplementary information.

Additional Editor Comments:

Thank you for the submission. Kindly address all points raised by the reviewers. In addition to the reviewer points, please ensure that the unit of observation in the study is clearly defined and inclusion criteria are clear. Please include defnitions of terms such as suppressed or non-suppressed viral load, WHO stages etc. In the discussion and limitations kindly discuss unmeasured factors that may confound results. Please comment on completeness of data and how missing data were handled. Please consider collinearity of variables included in the models e.g. ART regimen and duration, NVP prophylaxis and cotrimoxazole prophylaxis, as well as possible interactions.

Reviewers' comments:

Reviewer's Responses to Questions

**Comments to the Author**

1. Does this manuscript meet PLOS Global Public Health’s publication criteria?

Reviewer #1: Yes

Reviewer #2: Partly

2. Has the statistical analysis been performed appropriately and rigorously?

Reviewer #1: Yes

Reviewer #2: No

3. Have the authors made all data underlying the findings in their manuscript fully available (please refer to the Data Availability Statement at the start of the manuscript PDF file)?

Reviewer #1: Yes

Reviewer #2: No

4. Is the manuscript presented in an intelligible fashion and written in standard English?

Reviewer #1: Yes

Reviewer #2: Yes

Reviewer #1: Thank you for this meaningful study. There are few minor requires mostly relating to additional information:

1. For multivariable inclusion in your study, you noted a p >0.2 was included. Please could you give some feedback as to why that cut off was used?

2. There was quite a bit of time spent on the differences in outcomes between maternal age categories. Please could you describe why the ages were split into those specific ranges for the categorical variables and if you feel there is a specific characteristic unique to the age group selected (which is quite broad) that could account for the outcomes observed?

3. I understand that this was not done in the study but while it was meaningful to see the impact infant-related factors had on outcomes relative to maternal-related factors, it would be interesting to see if there is congruence or a relationship between the negative maternal factors and the negative infant factors which could have ended up compounding the subsequent effect or if the two categories (maternal and infant) truly were independent of each other.

Reviewer #2: Thank you for the opportunity to review this manuscript. It addresses an important area in national and global public health. Overall it was well written and relatively easy to follow. However there a number of omissions in the manuscript that the authors need to deal with. I have listed specific comments and areas that need to be addressed

Title- I think experience is preferable to perspective. Perspective had to do with how you see things or a point of view. But this is primary data analysis and think should be objective?

Abstract,

• Line 30: include dates of data collection and inclusion criteria

• Line 31- 32: infant diagnosis is PCR unless they are no longer breastfeeding. How old were this babies and at what point were the babies enrolled into the study? if diagnosis was PCR why call it serostatus. Maybe just HIV status

• Line 32- 33: descriptive statistics do not summarize the data; they describe the population included in your study

• Line 34: predictors of what?

• Line 37: describe the mother-infant pairs where there was transmission a bit more. Eg How old were the infants when final diagnosis was made

• Lines 43- 48: because the prevalence is very low, consider using a Poisson regression with robust error variance

Introduction

• Line 56: HIV what? Incidence/ new infections, mortality, prevalence?

• Line 64- 65: high viral load throughout pregnancy, delivery or breastfeeding is a key driver. Especially in the post-partum period

Methods

• Line 90: the study design should be retrospective cohort study based medical record or chart review. If the design was a retrospective cohort, was there a reason why survival analysis techniques were not used

• Line 94- 95: in the chart (which is not labelled) the authors state the study population was infants with a final result + maternal information available but hear it says infants and pregnant/ BF mothers Be consistent with the terms

• Line 96: the relevant Ugandan PMTCT guidelines should be summarized here or included as a supplementary file

• Line 97: were there other inclusion criteria eg infants born in a given period and followed up until a given time

• Line 109: the inclusion criteria in the middle box/ tier should be included with others in the section on study population., Who are the participants - infants?

• Line 115: regarding Stata 16 add manufacturer and city

• Line 116: regarding infant seroconversion, babies are born seropositive from exposure to maternal antibodies. Depending on age, they may not seroconvert but remain seropositive when infected. Please add in the setting section how infant testing is done in Uganda

Results

• Line 136: please describe the infants better before reporting on those who were HIV positive at the final diagnosis

• Table 2 lists under maternal characteristics viral suppression. At what time point since viral load is a time varying variable

• Table 3: there were only 18 babies that were HIV positive. The authors shouldn’t have more than two variables their models. Also data collection methods should include how variables were selected in the multivariate model

• Table 3: the authors should consider a Poisson regression with robust error variance

• Table 3: ART history of transition was not defined in the table footer

• Line 173: why compare the two models. It’s not either/ or. Did you try to build a model that combines both maternal and infant factors in one model

Discussion

• Line 196: most studies show higher MTCT risk with younger mothers. Can you explain reasons why this is not the case in your study? What % of babies had a final result? Did you compare babies with a final result and those without? Do these babies differ by maternal age? Please compare include a table before ta

• Line 239: the conclusion not to scale up PI based regimens in pregnancy doesn’t make sense. They are mostly used as third line or first or second line when DTG or EFV cannot be used. If the PIs need to be used what then is the recommendation

**Do you want your identity to be public for this peer review?** For information about this choice, including consent withdrawal, please see our Privacy Policy

Reviewer #1: No

Reviewer #2: No

---

## [Decision Letter · Decision Letter 1]

3 Nov 2025

PGPH-D-25-01533R1

Infant HIV-protection: Comparing Antiretroviral therapy, Maternal and Infant Factors Influence on Infant HIV acquisition in Uganda: A six-year Real-World Experience

Dear Dr. Ankunda

Thank you for submitting your manuscript to PLOS Global Public Health. After careful consideration, we feel that it has merit but does not fully meet PLOS Global Public Health’s publication criteria as it currently stands. Therefore, we invite you to submit a revised version of the manuscript that addresses the points raised during the review process.

Kindly address the reviewer comments below and resubmit.

Thank you for the opportunity to review the revised manuscript. It reads better than the previous version. The authors have addressed most comments but there are others that have not been addressed. I have listed the issue that need to be resolved below

- the abstract on the system and the one in the text are not the same. I don't know which one is correct

- although the authors changed the population to infants, the paper is still written like the mothers are the study population. eg i) Line 90-study design says study of medical records PLHIV attending PMTCT, ii) line 99- start with infant data since the infants are your participants and then describe how matching maternal data was collected, iii) Tables 1 and 3 where maternal data is presented.

the authors should present infant data first

- in the supplementary tables 1 and 2 shouldn't the denominator for the proportions be the number of HIV positive children and not those tested?

We look forward to receiving your revised manuscript.

Kind regards,

Nisha Anne Sunny Jacob, MBChB, M.Med, FCPHM (SA), PhD

Guest Editor

Journal Requirements:

Additional Editor Comments (if provided):

Reviewers' comments:

Reviewer's Responses to Questions

**Comments to the Author**

Reviewer #2: (No Response)

publication criteria?

Reviewer #2: Yes

3. Has the statistical analysis been performed appropriately and rigorously?

Reviewer #2: Yes

4. Have the authors made all data underlying the findings in their manuscript fully available (please refer to the Data Availability Statement at the start of the manuscript PDF file)?

Reviewer #2: No

5. Is the manuscript presented in an intelligible fashion and written in standard English?

Reviewer #2: Yes

Reviewer #2: Thank you for the opportunity to review the revised manuscript. It reads better than the previous version. The authors have addressed most comments but there are others that have not been addressed. I have listed the issue that need to be resolved below

- the abstract on the system and the one in the text are not the same. I don't know which one is correct

- although the authors changed the population to infants, the paper is still written like the mothers are the study population. eg i) Line 90-study design says study of medical records PLHIV attending PMTCT, ii) line 99- start with infant data since the infants are your participants and then describe how matching maternal data was collected, iii) Tables 1 and 3 where maternal data is presented.

the authors should present infant data first

- in the supplementary tables 1 and 2 shouldn't the denominator for the proportions be the number of HIV positive children and not those tested?

**Do you want your identity to be public for this peer review?** For information about this choice, including consent withdrawal, please see our Privacy Policy

Reviewer #2: No

---

## [Decision Letter · Decision Letter 2]

7 Dec 2025

PGPH-D-25-01533R2

Infant HIV-protection: Comparing Antiretroviral therapy, Maternal and Infant Factors Influence on Infant HIV acquisition in Uganda: A six-year Real-World Experience

Dear Dr. Akunda

Thank you for submitting your manuscript to PLOS Global Public Health. After careful consideration, we feel that it has merit but does not fully meet PLOS Global Public Health’s publication criteria as it currently stands. Therefore, we invite you to submit a revised version of the manuscript that addresses the points raised during the review process.

We look forward to receiving your revised manuscript.

Kind regards,

Nisha Anne Sunny Jacob, MBChB, M.Med, FCPHM (SA), PhD

Guest Editor

Journal Requirements:

Additional Editor Comments (if provided):

Thank you for the revision. Concerns about the statistical analysis have been raised by reviewers. Kindly review the detailed comments by reviewers and address accordingly. In addition to this, please note that the terminology of MTCT (mother-to-child-transmission) is changing to vertical transmission (VT) so that language is more inclusive and less stigmatising. Kindly edit the paper accordingly. Please also ensure the paper undergoes careful editing for grammatical, formatting and spelling errors including missing words, repeated words, inconsistent font etc.

Reviewers' comments:

Reviewer's Responses to Questions

**Comments to the Author**

Reviewer #2: (No Response)

Reviewer #3: All comments have been addressed

publication criteria?

Reviewer #2: Partly

Reviewer #3: Partly

3. Has the statistical analysis been performed appropriately and rigorously?

Reviewer #2: No

Reviewer #3: No

4. Have the authors made all data underlying the findings in their manuscript fully available (please refer to the Data Availability Statement at the start of the manuscript PDF file)?

Reviewer #2: Yes

Reviewer #3: Yes

5. Is the manuscript presented in an intelligible fashion and written in standard English?

Reviewer #2: (No Response)

Reviewer #3: Yes

Reviewer #2: Thank you for the opportunity to re-review this manuscript with the data included. A few more concerns

- line 35 still refers to mother-infant pairs when it should be infants

- Line 39 and 40. are these ORs for NVP and CTX adjusted ORs or crude ones. This is because table 3 shows different numbers

- Line 165 : the association between remaining on a PI based regimen was not significant in the multivariable model. Please indicate that this is a crude OR

- Table 3: did the univariable and multivariable logistic regression only include infants for whom maternal data was available? if that was the case, the regression model would have 18 outcomes and there only a maximum of two variables included. Its not indicated in the table how many infants were included in the infant model or maternal model

Reviewer #3: Manuscript Title: Infant HIV-protection: Comparing Antiretroviral Therapy, Maternal and Infant Factors Influence on Infant HIV Acquisition in Uganda

Manuscript ID: PGPH-D-25-01533R2

Reviewer Focus: Statistical and Methodological Integrity

Study Design

• The retrospective cohort design is appropriate for exploring associations between maternal and infant factors and HIV outcomes in real-world PMTCT settings. Six years of data collection adds temporal breadth. However, no a-priori power calculation or event-per-variable justification is provided. With only 18 positive outcomes (1.96%), the analysis is underpowered. The retrospective nature precludes control for unmeasured confounding. Exclusion criteria were not clearly defined, introducing selection bias.

• Recommendation: Include a post-hoc power analysis and clearly define exclusion logic. Frame findings as exploratory rather than confirmatory.

Variable Definition and Data Handling

• No. clear description of how missing data was handled. No testing of missingness mechanism (MCAR, MAR, MNAR) was performed.

• How was adherence measured from retrospective data?

• Variable selection for the multivariable model is unclear. `or example the exclusion of maternal age and adherence but the inclusion of history of transition despite evidence from the bivariate/univariate models.

• Recommendation: Apply multiple imputation or sensitivity analysis for missingness.

Model Specification and Validity

• The term 'hierarchical logistic regression' is misused; no multilevel structure seems to be described or shown. It looks like two independent logistic models were run.

• With 18 HIV-positive cases and >8 covariates, event-per-variable ratio ≈ 2, causing overfitting. A more robust regression model (mixed effects or penalized is preferred). It is not clear how you handled multiple births/ infants from one mother.

• No model calibration or discrimination metrics reported.

Model Comparison (AIC/BIC)

• AIC/BIC values were compared across models with different sample sizes (821 vs. 962), which invalidates comparability. Interpretation of AIC as model superiority is incorrect.

Recommendation: Use likelihood-ratio tests or cross-validated AUCs on identical datasets for valid comparison.

Statistical Reporting and Transparency

• Extremely wide CIs indicate quasi-complete separation. No pseudo-R² reported. No multiple testing correction.

• Recommendation: Report Nagelkerke R², apply Bonferroni/FDR corrections, and identify unstable estimates explicitly.

Rare-Event Bias

• Standard logistic regression under rare events yields biased MLE estimates.

• Recommendation: Implement rare-event logistic regression or penalized likelihood methods. Also explains the very wide Cis.

Interpretation and Validity of Conclusions

• Claims of 'better model fit' are overstated; evidence supports exploratory correlation, not predictive superiority.

• Recommendation: Revise language to reflect exploratory, hypothesis-generating interpretation.

Reviewer Summary Statement

The study is conceptually valuable and clinically relevant but suffers from significant statistical fragility due to rare events, over-fitting, and improper model comparison. Re-analysis with penalized regression, improved model diagnostics, and reproducible code is required before acceptance for publication.

**Do you want your identity to be public for this peer review?** For information about this choice, including consent withdrawal, please see our Privacy Policy

Reviewer #2: No

Reviewer #3: No

---

## [Decision Letter · Decision Letter 3]

19 Jan 2026

PGPH-D-25-01533R3

Infant HIV-protection: Comparing Antiretroviral therapy, Maternal and Infant Factors Influence on Infant HIV acquisition in Uganda: A six-year Real-World Experience.

Dear Dr Dr Ankunda

Thank you for submitting your manuscript to PLOS Global Public Health. No responses were received for two of the points in the last review - see editor's comments below. As such the manuscript does not fully meet PLOS Global Public Health’s publication criteria as it currently stands. Therefore, we invite you to submit a revised version of the manuscript that addresses the points raised during the review process.

We look forward to receiving your revised manuscript.

Kind regards,

Nisha Anne Sunny Jacob, MBChB, M.Med, FCPHM (SA), PhD

Guest Editor

Journal Requirements:

Additional Editor Comments (if provided):

**Thank you for the revision. No responses were received for the following two points. Kindly address before acceptance for publication:**

1. Please note that the terminology of MTCT (mother-to-child-transmission) is changing to vertical transmission (VT) so that language is more inclusive and less stigmatising. Kindly edit the paper accordingly.

2. Please also ensure the paper undergoes careful editing for grammatical, formatting and spelling errors including missing words, repeated words and inconsistent font which were identified on review.

Reviewers' comments:

Reviewer's Responses to Questions

**Comments to the Author**

Reviewer #3: All comments have been addressed

publication criteria?

Reviewer #3: Yes

3. Has the statistical analysis been performed appropriately and rigorously?

Reviewer #3: Yes

4. Have the authors made all data underlying the findings in their manuscript fully available (please refer to the Data Availability Statement at the start of the manuscript PDF file)?

Reviewer #3: (No Response)

5. Is the manuscript presented in an intelligible fashion and written in standard English?

Reviewer #3: Yes

Reviewer #3: All comments have been sufficiently addressed.

**Do you want your identity to be public for this peer review?** For information about this choice, including consent withdrawal, please see our Privacy Policy

Reviewer #3: No

---

## [Editor Report · Decision Letter 4]

27 Jan 2026

Infant HIV-protection: Comparing Antiretroviral therapy, Maternal and Infant Factors Influence on Infant HIV acquisition in Uganda: A six-year Real-World Experience.

PGPH-D-25-01533R4

Dear Collins Ankunda

We are pleased to inform you that your manuscript 'Infant HIV-protection: Comparing Antiretroviral therapy, Maternal and Infant Factors Influence on Infant HIV acquisition in Uganda: A six-year Real-World Experience.' has been provisionally accepted for publication in PLOS Global Public Health.

Best regards,

Nisha Anne Sunny Jacob, MBChB, M.Med, FCPHM (SA), PhD

Guest Editor